# Uncovering the Mechanism of the Role of Fly Ash in the Self-Healing Ability of Mortar with Different Curing Ages

**DOI:** 10.3390/ma16093453

**Published:** 2023-04-28

**Authors:** Congqi Luan, Lianwang Yuan, Jinbang Wang, Zonghui Zhou

**Affiliations:** 1Shandong Provincial Key Laboratory of Preparation and Measurement of Building Materials, University of Jinan, Jinan 250022, China; 2Shandong Hi-Speed Road & Bridge Technology Co., Ltd., Jinan 250021, China

**Keywords:** self-healing, fly ash mortar, calcium carbonate precipitation

## Abstract

As an admixture of cement-based materials, the reaction of fly ash (FA) usually takes place in the late age of curing, so FA will affect the self-healing ability of long-age cement-based materials. The self-healing potential and the characteristics of self-healing products of cementitious materials before and after crack healing were analyzed by microscopic tests, and the mechanism of the effect of fly ash on the self-healing performance of cementitious materials was revealed. The results showed that the increase in fly ash content promoted the improvement of the self-repair performance of cracked specimens at 28 d, especially when the fly ash dosage was 40%, the crack opened after 30 d of healing in water was completely closed, the UPV value after recovery was close to 3000 m/s, the self-repair efficiency of maximum amplitude and main frequency amplitude was up to more than 60%, and the recovery rate of compressive strength was increased to more than 30%. However, the increase in fly ash content was not conducive to the self-repair of cracked samples at 210 d, and with the increase in fly ash content, the crack closure effect weakened, the UPV value after recovery decreased, the crack repair rate based on ultrasonic transmission decreased to about 20%, and the compressive strength recovery rate increased slightly. In addition, calcium carbonate precipitation was the main repair product of crack filling and healing, including calcite and spherulite. With the increase in fly ash content, the content of element C in the self-repair products of 28-day-old specimens gradually increased, and the size of calcium carbonate crystals gradually decreased, but the filling was denser, whereas the calcium carbonate crystals in the self-repair products of 210 d specimens gradually became fine and loose.

## 1. Introduction

Cracking is a primary cause of concrete damage and durability issues [1,2]. It facilitates the transportation of harmful agents that accelerate the corrosion of steel bars and damage concrete structures [3]. Therefore, cracks are potential hazards that must be repaired or filled in. Improving concrete’s self-healing ability provides a practical method for crack repair.

Concrete’s self-healing ability, also known as autogenous self-healing, occurs due to the further hydration of unhydrated cement particles and admixture materials [4]. During the early hydration stage, these particles are coated by C-S-H gels or other hydration products and are not fully hydrated [5]. When cracks and water appear, the unhydrated particles continue to hydrate or swell, filling the cracks [6,7]. Furthermore, the self-healing ability of concrete shows great potential for crack repair. 

To improve and obtain a remarkable self-healing ability in concrete, numerous studies have been conducted based on healing conditions and alternative cementitious materials. Yuan et al. [8,9] found that the self-sealing ratios in the static liquid water environment were higher than in other environments, including in air with various humidity, flowing water, and wet–dry cycles. They also noted that cement containing more 30–60 nm cement particles in mortar had a better self-healing ability in the early stage. Supplementary cementitious materials, including silica fume, slag, and FA, are used to increase concrete’s self-healing ability [10]. Youngcheol Choi et al. [11] showed that incorporating silica fume into concrete improved its self-healing ability and reduced the chloride ion penetration depth after crack healing. Zhou et al. [12] demonstrated that concrete containing 30% slag or 40% FA presented optimal self-healing ability. Similar results were obtained by Van Tittelboom et al. [13]. Furthermore, Mustafa Sahmaran et al. [14] found that engineered cementitious composite (ECC) samples with FA had a higher self-healing ability than those with slag. Guan et al. [15] investigated exposure conditions that had different influences on concrete’s self-healing ability containing limestone. They found that exposure to Ca(OH)_2_ solutions and the incorporation of limestone improved concrete’s self-healing ability. Li et al. [16] illustrated that crystalline admixture (CA) ensures self-healing properties, the mortar with CA and 10 wt.% granulated blast furnace slag has the highest self-healing capability, and the healing product is mainly composed of calcium carbonate. Qian et al. [17] studied the effect of pre-cracking time on concrete self-healing ability and demonstrated that the self-healing deflection capacity of pre-cracked samples at 14 and 28 d recovered or exceeded that of the original samples at almost all pre-cracked ages. Additionally, large volumes of fly ash were introduced into the cement to improve the performance of concrete and reduce crack width.

As a common cementitious material, fly ash plays a significant role in workability and long-age performance development in concrete. Due to its pozzolanic reaction, fly ash undergoes secondary hydration, consumes CH, and forms new C-S-H gel in the later stage, which changes the microstructure and impacts the self-healing ability of concrete. Pipat Termkhajornkit et al. [18] demonstrated that the fly ash–cement system has the self-healing ability for cracks that occur from shrinkage, and the self-healing ability increased when the fraction of fly ash increased. However, their research focuses on behavior after 28 d. Zhigang Zhang et al. [19] pointed out that the secondary hydration of fly ash in the later stage can change the microstructure of concrete and also show a positive impact on self-healing behavior. While self-healing with FA in concrete is available, the self-healing ability of mortar with FA after long-term curing is limited in the literature.

In this study, five groups of mortar samples were investigated for the self-healing ability of long-age-cured mortar incorporating different FA contents. Therefore, mortar crack closure was observed by the optical microscope and evaluated by the ultrasonic pulse velocity (UPV). In addition, the ultrasonic waveform and frequency further demonstrated the accuracy of the crack self-healing ratio. Moreover, the composition and micromorphology of products were determined by XRD, TG, and SEM-EDS to characterize the mechanisms of the self-healing ability of mortar. 

## 2. Materials and Methods

### 2.1. Materials 

In this paper, Portland cement (P·O 52.5) was bought from Shanshui Co., Ltd., and the P·O 42.5 and FA were supplied by the Shandong Lubi Co., Ltd. Accordingly, GB/T176-2017 [20], the chemical composition of FA and cement were analyzed by X-ray fluorescence spectrometer (S8 TIGER, Brook). First, a small amount of cement or fly ash powder was pressed into sheets of about 28 mm and placed into the instrument. When the chemical elements in the sample were excited by the high-energy radiation emitted by the X-ray tube, they emitted characteristic X-rays. Calculate the relationship between the mass and the X-ray intensity of an element in the sample through X-ray fluorescence spectrometer, and convert the corresponding element mass percentage, which is listed in Table 1. The XRD pattern and particle size of P·O 42.5 and FA are presented in Figure 1 and Figure 2, respectively. The fineness modulus and maximum size of river sand were 2.8 and 4.75 mm, respectively. Put the sieves of different particle sizes, from small to large, in order. Put a 500 g sieve into the largest aperture sieve, put it on the vibrating sieve machine (ZBSX-B2A, Hebwi Shuangxin Co., Ltd.) after vibration, and calculate the percentage of sieving on each level of the sieve. The characteristics of river sand are presented in Table 2. The water-to-binding materials ratio and the binding materials-to-river sand ratio of mortar were fixed at 0.4 and 1:3, respectively. The mixed proportion of mortar was shown in Table 3.

### 2.2. Samples Preparation 

After mixing the cement paste for 30 s, the sand was added evenly within 30 s. The mortar was mixed at a slow rate for 120 s and then at a high rate for 90 s. The 40 mm × 40 mm × 160 mm prismatic mortar specimens and ϕ100 × 50 mm cylindrical mortar specimens were cast and then covered by a plastic film for 1 d curing in the standard chamber. The samples were demolded after one day of curing and cured for 28 d, 90 d, 210 d, and 300 d in standard conditions (25 ± 2 °C, 95% RH).

### 2.3. Experimental Procedures

#### 2.3.1. Prefabricating and Measurement Cracks and Compressive Strength

Figure 3 illustrates the schematic diagram of a single crack in a cylinder mortar specimen, which was produced through a split test with a loading rate of 100 N/s. Microcracks were created in the mortar using a steel bar while ensuring that the mortar remained intact. The cylinder mortar was then fixed with epoxy resin and a rubber tube to restrict the expansion of cracks during the self-healing process. To test the recovery of compressive strength, distributed microcracks were induced in the cubic mortar by preloading damage until the ultimate compressive strength was reached, and then the loading was immediately halted to maintain the integrity of the pre-damaged mortar.

In this study, the pT-C10 crack width gauge was utilized to monitor the changes in the width of a single surface crack after pre-cracking and self-healing. Three equally spaced fracture observation points were selected and marked, and the crack width was measured and recorded using specialized measurement software.

The compressive strength result of the mortar was obtained from six specimens with a loading rate of 2.4 kN/s. 

#### 2.3.2. Ultrasonic Tests

An ultrasonic pulse velocity (UPV) test has been widely used in concrete structure detection, and reliable results have been obtained in the characterization of crack healing. UPV and transmitted ultrasonic wave test instruments and their mechanism are shown in Figure 4. Before the test, the mortar was dried at 50 °C for 24 h, eliminating the absorption of ultrasonic energy by free water. The device was provided by Shenborui Company, China. The natural frequency of sensors was 54 kHz, the pulse voltage was 1 kV, and the resolution was 0.1 μs. The transmitted ultrasonic frequency was 150 kHz, which was sent out by the pulse transmitter (Tektronix AFG 3022C) and received by the mixed domain oscilloscope (Tektronix MDO 3024). Moreover, some measures were conducted to decrease the influence of external factors: First, all ultrasonic tests were completed in interior environments. Second, the surface contacted with transducers was polished with 1000 # abrasive paper. The thickness loss could be neglected. Vaseline was applied on the interface evenly to enhance the coupling of the transducer and the specimen. Third, the transducer was at the same position in each ultrasonic test. The loss rate was calculated to determine the self-healing rate. In addition, the frequency data were obtained by the fast Fourier transform. In a PUV test, the side faces of the transducer and the specimen were aligned. The distance between the two sensors was determined as 110 mm. In an ultrasonic wave test, the sensors were at the two ends of the specimen. The distance was 160 mm. Although the amplitude value would slightly change in a repeated test of one specimen, the amplitude tendency of uncracked, cracked, and repaired specimens is regular and repeatable. 

#### 2.3.3. Characterization of Cement Past and Self-Healing Products

The hydration products of paste and self-healing were collected, dried, and broken into powder for XRD (Brook D8 Advance, Germany) and TG tests. The powder was heated from 25 °C to 1000 °C at 10 °C/min in an argon atmosphere for TG and scanned from 5° to 80° with a step size of 0.02° for XRD analysis. The specimen was dried at 60 °C for 24 h and then covered by gold to wait for the detection. The microscopic morphology of self-healing products with different curing ages was observed by the SEM-EDS (ZeissEVO LS 15, Germany). 

#### 2.3.4. Mechanical Strength

On 7 d, 28 d, 210 d, and 300 d, the flexural and compressive strength were tested according to GB/T 17617-1999 (Chinese standard) [21].

## 3. Results and Discussion 

### 3.1. Effect of Fly Ash on Mortar Mechanical Properties 

Effect of fly ash content on mortar mechanical properties is displayed in Figure 5. Due to the pozzolanic reaction characteristics, the consequence of fly ash on mortar mechanical strength in the early and late stages was the opposite. With fly ash increases, the early flexural and compressive strength of mortar decreased slowly. Because the fly ash replaced cement, the quality of cement decreased, resulting in a decreased amount of hydration products. In addition, the low activity of fly ash in the early stage produced few hydration products and could not meet its dilution effect, so the early strength of mortar decreased [22,23]. With the curing age increasing, fly ash reacted with CH and produced new hydration products, which filled the pores and the microstructure and improved the mechanical properties of mortar. At 28 d, the group FA-2 mechanical strength containing 20% FA had a little higher than that of the control. After curing for 210 d, the strength of the mortar gradually increased with the increase in fly ash, which indicated that the fly ash pozzolanic reaction had been carried out and further hydration of cement was promoted, resulting in the density of matrix and the late strength being further increased. 

The effect of fly ash content on the mechanical properties of mortar is illustrated in Figure 5. The pozzolanic reaction characteristics of fly ash had opposite effects on the early- and late-stage mechanical strength of the mortar. As the fly ash content increased, the early flexural and compressive strength of the mortar decreased gradually. This was because the fly ash replaced cement, resulting in a decrease in the quality of cement and a reduced amount of hydration products. Additionally, the low activity of fly ash in the early stage produced only a small amount of hydration products, which were insufficient to compensate for its dilution effect, leading to a decrease in the early strength of the mortar [22,23]. However, with the curing age increasing, fly ash reacted with CH and produced new hydration products that filled the pores and microstructure, thereby improving the mechanical properties of the mortar. At 28 d, the group FA-2 containing 20% fly ash exhibited slightly higher mechanical strength than the control. After 210 d of curing, the strength of the mortar gradually increased with an increase in fly ash content, indicating that the pozzolanic reaction of fly ash had occurred, and further hydration of cement was promoted, resulting in an increased density in the matrix and improved late strength.

### 3.2. Effect of FA on Self-Healing Properties of Mortar

#### 3.2.1. Observation of Surface Crack 

The study utilized an optical microscope to directly observe and measure the surface crack width of mortar, providing an accurate evaluation of the self-sealing effect. Prefabricated cracks were induced in the mortar specimens, which were then allowed to self-heal for 30 d and 90 d after curing for 28 d and 210 d, respectively. The influence of fly ash content on the closure of mortar cracks with different curing ages is displayed in Figure 6. Figure 6a illustrates the change in crack width after 30 d of self-healing in water for mortar cured for 28 d. The initial average widths of cracks for mortars containing 0%, 10%, 20%, 30%, and 40% fly ash were 111 μm, 104 μm, 149 μm, 200 μm, and 279 μm, respectively. Mortar containing 40% fly ash showed complete closure of cracks after 30 d of self-healing in water. The degree of crack self-healing improved with an increase in fly ash content for short-term curing (28 d). Figure 6b displays the variation in crack width for mortar cured for 210 d. The closure of cracks decreased with an increase in fly ash content, which is the opposite result compared to that for 28-day curing. After 90 d of self-healing in water, the cracks in the control group were almost closed due to the settlement of sufficient Ca^2+^ at the fracture site after a long period of diffusion and reaction [17]. Due to the pozzolanic reaction, fly ash consumed a large amount of calcium hydroxide, which reduced the Ca^2+^ content used for CaCO_3_ precipitation [22,23]. Furthermore, the further hydration of cement and the fly ash pozzolanic reaction increased the density of the mortar and reduced porosity, limiting the dissolution of Ca^2+^ calcium ions [24].

#### 3.2.2. Ultrasonic Tests

The ultrasonic pulse velocity (UPV) is a reliable method to assess the compaction and homogeneity of concrete [25,26]. The UPV value of concrete increases with its higher compaction and homogeneity. The impact of fly ash content on the UPV of cracked mortar specimens cured for 28 d (a) and 210 d (b) is presented in Figure 7. For uncracked mortar, the UPV values were high and followed a similar pattern as that of the strength. However, when the prefabricated crack was introduced, the UPV values dropped significantly [27], as indicated by the red line in Figure 7. After the self-healing process for 30 d, the UPV values of 28-day cured mortar increased with the addition of fly ash, indicating that FA improved the early age crack self-healing ability. Conversely, in the case of 210-day cured mortar, despite the 90-day self-healing in water, the addition of fly ash reduced the self-healing ability. The UPV values of the self-healed mortar were consistent with the closure of surface cracks.

A transmission ultrasonic wave was used to characterize the seal-healing of cracks in the mortar. The healing rate characterized the healing effect better. Figure 8 and Figure 9 present the influence of FA on the healing ratio of maximum amplitude (UM) and frequency amplitude (UF) at 28 d and 210 d, respectively. The UM and UF healing ratios were calculated from Formulas (1) and (2), respectively.
(1)$$\mathrm{UM}\;\mathrm{healing}\;\mathrm{rate}=\left[1-\left(M_{u}-M_{s}\right)/\left(M_{u}-M_{c}\right)\right]\times 100\%$$
(2)$$\mathrm{UF}\;\mathrm{UF}\;\mathrm{healing}\;\mathrm{rate}=\left[1-\left(F_{u}-F_{s}\right)/\left(F_{u}-F_{c}\right)\right]\times 100\%$$
where *M* and *F* represented the maximum amplitude of the ultrasonic wave and frequency amplitude of samples. *u*, *c*, and *s* represent uncracked samples, cracked samples, and seal-healed samples, respectively. 

The UM and UF of the transmitted ultrasonic waves were affected by the densification of the hardened mortar and the filling effect inside the crack. The existence of cracks greatly weakened the ultrasonic transmission energy, which reduced the values of amplitude and frequency [28,29]. With the filling and seal-healing of cracks, the ultrasonic transmission energy increased, and then amplitude and frequency recovered [30], which indicated that the better the crack seal-healing effect was, the larger the amplitude and the higher the frequency were. Therefore, the cracks of 28 d mortar healed better with fly ash increases, resulting in the maximum amplitude and the amplitude of the main frequency increasing, which was attributed to the increased deposition of calcium carbonate precipitation in cracks. However, the cracks of the 210 d mortar seal-healing became weakened with an increase in fly ash. After long-age curing, the hydration degree of the mortar was further added, which increased the densification of the mortar and limited the dissolution of Ca^2+^. In addition, a large amount of fly ash consumed the calcium hydroxide and formed stable hydration products because of the pozzolanic reaction. Both weakened the concentration of Ca^2+^ in the solution and reduced the deposition of CaCO_3_ gathering around the edge of the crack.

#### 3.2.3. Strength Recovery 

The effect of fly ash on the recovery of strength in cracked samples cured for 28 d and 210 d is shown in Figure 10. The healing ratios of strength were calculated using the formulas:(3)$$\mathrm{Healing}\;\mathrm{ratio}\;\mathrm{of}\;\mathrm{strength}=\left[\left(F_{m}-F_{n}\right)/\left(F_{m+n}-F_{n}\right)\right]\times 100\%$$
where *F_m_* represents the compressive strength of the mortar after healing for *m* d, *F_n_* represents the compressive strength of the mortar after curing for *n* d, and *F_m_*_+*n*_ represents the compressive strength of the mortar after curing for *m* + *n* d. Both 28-day and 210-day cured mortars with prefabricated cracks showed that the healing ratio of compressive strength increased with an increase in fly ash after a certain period of healing in water. Compared with the results of crack closure and ultrasonic recovery, the healing ratio of 28-day compressive strength showed a similar pattern, while reverse results were observed for the 210-day cured mortars.

Both cured 28 d and 210 d mortars with prefabricated cracks showed the same rule that the healing ratio of compressive strength increased with fly ash increases after a certain period of healing in water. This is because continued hydration and calcium carbonate precipitation had different roles in different characterization methods. For crack closure and ultrasonic recovery, calcium carbonate precipitation was the major contributor to filling the cracks, but it contributed less than continued hydration in the healing ratio of compressive strength. With an increase in fly ash, the continued hydration effect was enhanced, which led to the continued hydration products repairing the cracks and improving the strength [27]. The healing ratio of 28-day mortar was generally higher than that of 210-day mortar as the further hydration of cement and fly ash was limited at an early age, and then they continued to hydrate and harden during self-healing in water, thus significantly improving the compressive strength [31].

However, after curing for 210 d, the high hydration degree of cement reduced the ability of continued hydration, resulting in a negative healing rate including limited improvement by the pozzolanic reaction of fly ash [7,32]. Moreover, the microcracks produced by the pre-pressure damage were much smaller than 300 μm, and it is difficult for CO_2_ to enter this microcrack, which is the reason why the continued hydration products can fill and play a major role.

#### 3.2.4. Correlation Analysis

Figure 11 shows the correlation between fly ash content and self-healing ratios. In the case of the 28 d mortar, the self-healing ratio increased as the fly ash content increased. This was due to the deposition of calcium carbonate at the edges of cracks, resulting in a higher crack closure rate compared to the UPV self-healing rate, which was lower due to the less dense surface of the crack than the matrix. However, the autogenous self-healing ability had a limited effect on the mechanical properties improvement, resulting in a relatively low self-healing ratio of compressive strength.

For 210 d mortar, the crack closure ratio decreased as the fly ash content increased, as there was a decrease in calcium carbonate precipitation content. This was due to the reaction between fly ash and Ca(OH)_2_, leading to the consumption of Ca(OH)_2_ and a lower release of Ca^2+^ during the hydration process of fly ash. On the other hand, the strength recovery ratio increased with fly ash content due to the filling and cementation of microcracks by the reaction products from the pozzolanic reaction of fly ash. The recovery mechanism of the crack healing ratio was different from that of the strength recovery ratio. The surface crack closure healing was mainly contributed by calcite precipitation near the crack opening, whereas the strength recovery was dominated by the formation of the C-S-H gel phase [31]. The addition of fly ash improved the self-healing performance of the mortar, particularly in the recovery of strength through the pozzolanic reaction of fly ash. When the crack was produced in the early hydration age (within 28 d), the fly ash was conducive to the precipitation of calcium carbonate by promoting the hydration of cement. However, the addition of fly ash inhibited the precipitation of CaCO_3_ and had a negative impact on the healing of cracks in 210 d samples.

### 3.3. Self-Healing Mechanism Analysis of Mortar Based on Fly Ash Content

#### 3.3.1. Self-Healing Potential Analysis of Mortar Based on Fly Ash Content 

Figure 12 presents an XRD analysis of initial crack surface hydration products for samples cured 28 d (a) and 210 d (a). In the 28 d mortar, the diffraction peaks of hydration products, calcium hydroxide, and unreacted cement clinker minerals were observed. The intensity of the diffraction peak of calcium hydroxide did not show significant changes with the addition of fly ash, indicating no pozzolanic reaction occurred with fly ash after 28 d of curing. The presence of large amounts of calcium hydroxide and unreacted fly ash in the mortar provided a great potential for self-healing. In the case of the 210 d mortar, the diffraction peak strength of calcium hydroxide and cement minerals decreased significantly, indicating further hydration of cement and the pozzolanic reaction between calcium hydroxide and fly ash, leading to the formation of new C-S-H gel and an increase in mortar strength. With the increase in fly ash content, the diffraction peak of calcium hydroxide decreased gradually, indicating a higher degree of pozzolanic reaction, which contributed to the development of strength and the densification of the microstructure. However, the consumption of calcium hydroxide by the pozzolanic reaction of fly ash reduced the precipitation of calcium carbonate and hindered the autogenous self-healing of the mortar. 

The TG analysis of the initial crack surface material provided quantitative evidence for the fly ash pozzolanic reaction and calcium carbonate precipitation. As shown in Figure 13, the decomposition of ettringite and C-S-H gel at 70~240 °C indicated that the amount of hydration products increased with the extension of curing time. The content of calcium hydroxide decreased with the increase in fly ash but for different reasons. In the case of the cured 28 d samples, the increase in fly ash caused a decrease in the relative cement content and the production amount of calcium hydroxide. In the case of the cured 210 d samples, a large amount of Ca(OH)_2_ was consumed by the fly ash pozzolanic reaction. These findings were consistent with the results of the self-healing ratio and strength recovery ratio. The study showed that the strength development of mortar in the early stage and the long age exhibited different rules with the change in fly ash content, which was consistent with the crack-healing behavior. Specifically, the cracks were completely closed for the 28 d cured mortar and not closed for the 210 d cured mortar.

#### 3.3.2. Analysis of Self-Healing Products of Mortar Based on FA Content

Figure 14 presents an XRD analysis of self-healing products in cracks produced in samples cured for 28 d and 210 d. The main self-healing products deposited were CaCO_3_, including calcite and vaterite. The precipitation of two crystal types of CaCO_3_ was related to the pH value of the solution. For the control sample without fly ash, the pH value increased rapidly in the self-healing process due to the high content of calcium hydroxide which promoted the precipitation of vaterite. With the progress of the reaction, the pH value decreased slightly, and the calcite precipitation increased. Therefore, the intensity of the vaterite diffraction peak decreased with the increase in fly ash.

Figure 15 shows the TG analysis of self-healing products in cracks produced in samples cured for 28 d and 210 d. For self-healing products obtained from the mortar cured for 28 d, the TG curve shows that the significant weight loss peak is concentrated between 600 °C and 840 °C due to the decomposition of calcite. The CaCO_3_ content increased with the increase in fly ash, which expressed a reversed rule in the calcium hydroxide content of the crack surface before healing as the fly ash promoted the hydration of cement. The activated alumina and amorphous silica in the fly ash reacted with the calcium hydroxide, which consumed part of the calcium hydroxide and promoted the reaction of the unhydrated cement particles to produce calcium hydroxide. In addition, the increase in the calcium hydroxide content promoted the further hydration of fly ash in the later stage [2]. The calcium hydroxide content increased with fly ash increase. This was because part of the cement was replaced by fly ash, which diluted the cement and led to the actual higher w/c than that of the theory, which enhanced the further hydration degree of the cement and fly ash and increased the content of calcium hydroxide. Therefore, the continued hydration of cement released more calcium ions and increased the production amount of calcium carbonate during self-healing. 

For the samples cured for 210 d, the self-healing products were also calcium carbonate. However, the production of calcium carbonate gradually decreased with the increasing content of fly ash, which was consistent with the change rule of the calcium hydroxide content on the crack surface before healing. This indicated that insufficient Ca^2+^ reduced the deposition of calcium carbonate in the self-healing process, which was affected by the high hydration degree of the cement and the fly ash pozzolanic reaction. It was further confirmed that the main product of crack filling and healing was depositional calcium carbonate, which was not only related to the content of calcium hydroxide but also related to the Ca^2+^ dissolution of the continued hydration of cement in the process of self-healing [33]. The high pozzolanic reaction of fly ash had a low contribution to the crack healing. 

SEM-EDS analysis of self-healing products in cracks produced in samples cured for 28 d is shown in Figure 16. By comparing five SEM images, it can be found that in Figure 16-KB (control sample), the full, well-developed, massive morphology of calcium carbonate crystal was observed, but with the increase in fly ash content, calcium carbonate crystal presented a more homogenous and fine size as shown in Figure 16. FA-4. EDS dots demonstrated that the content of elemental carbon in the crystal gradually increased with fly ash increasing, indicating that the content of calcium carbonate also gradually increased. This was consistent with TG analysis and proved that the increase in fly ash had a better self-healing ability for mortar at 28 d. 

SEM-EDS analysis of self-healing products at cracks produced in samples cured for 210 d is shown in Figure 17. SEM images showed that the coarse and rod-shaped calcium carbonate crystals became loose and fine with fly ash increases, indicating that increasing the content of fly ash was unfavorable to the self-healing effect in the mortar. In addition, Figure 17 (KS) illustrates that completely developed calcium carbonate crystals are rod-like or cluster-shaped, reducing the compactness of the microstructure, which explained the reason for the decrease in strength after 90 d of self-healing in water for the 210 d aged mortar in Figure 10b. By comparing Figure 17 with Figure 16, silicon was observed in the 210 d self-healing products due to the failure of calcium carbonate deposition in covering the original hydration products, indicating that the long-aged self-healing ability was gradually weakened.

#### Summaries on Required Conditions for Different Mechanisms of Self-Healing

Table 4 provides a summary and comparison of the cracking and strength recovery achieved through different self-healing mechanisms. Most of the adhesive agents used for self-healing in cementitious materials cannot be hardened in water. By now, almost all types of bacteria explored for self-healing in cementitious materials are ones that can induce or promote the formation of CaCO_3_. For self-healing based on bacteria, the presence of ${\mathrm{CO}}_{3}^{2-}$ ions is necessary. For autogenous self-healing, the presence of ${\mathrm{CO}}_{3}^{2-}$ ions can enhance the healing efficiency when portlandite in cracks is carbonated. It was reported that by means of autogenous self-healing, only cracks with a width of less than 50 μm can be healed. In this study, the use of 40% fly ash in mortar resulted in cross-strength crack healing and strength recovery after 30 d. The fly ash was able to effectively heal cracks up to 270 μm with good strength recovery and was found to be comparable in performance to self-healing concrete incorporating mineral admixtures or bacteria. Overall, the results suggest that the use of fly ash has significant potential for improving the self-healing capabilities of concrete. The self-healing ability of fly ash is not weaker than that of mineral admixtures, bacteria, or adhesives.

## 4. Conclusions

1. The increase in fly ash content promoted the improvement of the self-repair performance of cracked specimens at 28 d, especially when the fly ash dosage was 40%, the crack opening after 30 d of healing in water was completely closed, the UPV value after recovery was close to 3000 m/s, the self-repair efficiency of maximum amplitude and main frequency amplitude was up to more than 60%, and the recovery rate of compressive strength was increased to more than 30%. However, the increase in fly ash content was not conducive to the self-repair of cracked samples at 210 d, and with the increase in fly ash content, the crack closure effect weakened, the UPV value after recovery decreased, the crack repair rate based on ultrasonic transmission decreased to about 20%, and the compressive strength recovery rate increased slightly. The pozzolanic reaction of fly ash improved the self-repair performance of cementitious materials, especially the strength recovery; however, the absence of Ca^2+^ in fly ash inhibited the precipitation of calcium carbonate, which was not conducive to crack filling.

2. The initial crack surface of the 28 d samples contained calcium hydroxide and unreacted cement clinker minerals, which provided great potential for crack self-repair, whereas the crack surface of the 210 d samples contained less calcium hydroxide and unhydrated cement clinker minerals, and the intensity of the diffraction peak of calcium hydroxide gradually decreased with the increase in fly ash admixture and the content decreased.

3. Calcium carbonate precipitation was the main repair product of crack filling and healing, including calcite and spherulite. With the decrease in fly ash admixture, the content of calcium hydroxide increased, which promoted the precipitation of spherulite. With the reaction of fly ash, the precipitation of calcite increased. With the increase in fly ash content, the content of element C in the self-repair products of 28-day-old specimens gradually increased, and the size of calcium carbonate crystals gradually decreased, but the filling was denser, whereas the calcium carbonate crystals in the self-repair products of 210 d specimens gradually became fine and loose.

## Figures and Tables

**Figure 1 materials-16-03453-f001:**
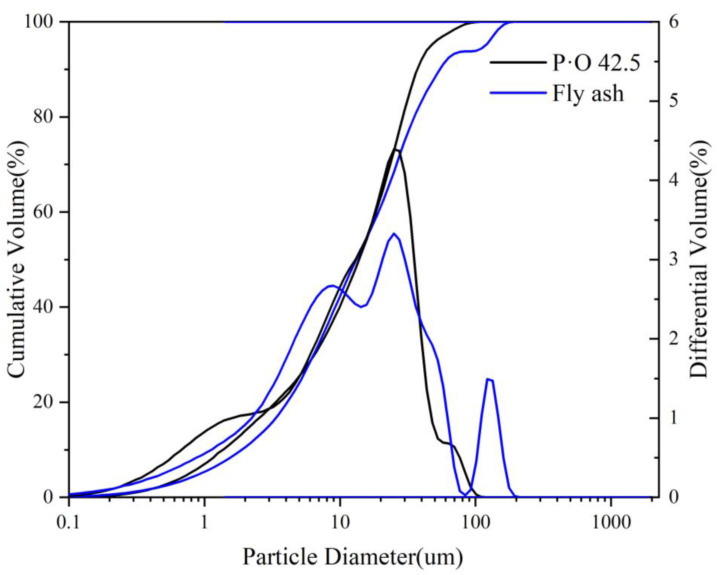
Particle size of P·O 42.5 and fly ash.

**Figure 2 materials-16-03453-f002:**
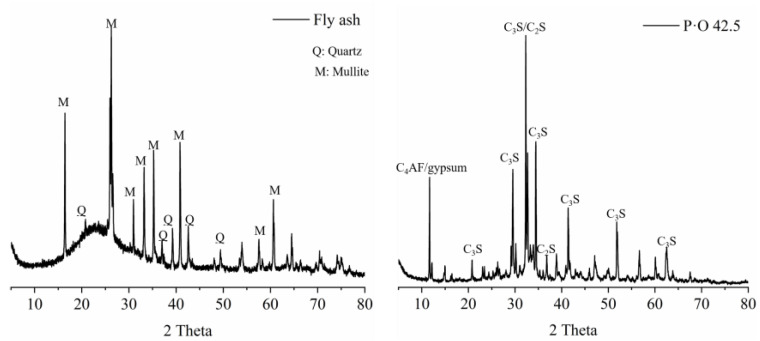
XRD pattern of fly ash and P·O 42.5.

**Figure 3 materials-16-03453-f003:**
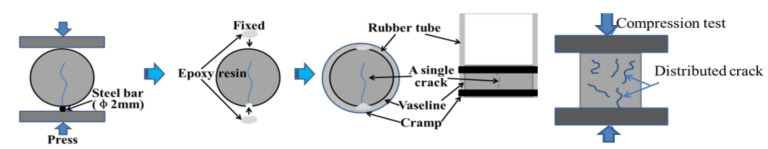
The schematic diagram of a single crack.

**Figure 4 materials-16-03453-f004:**
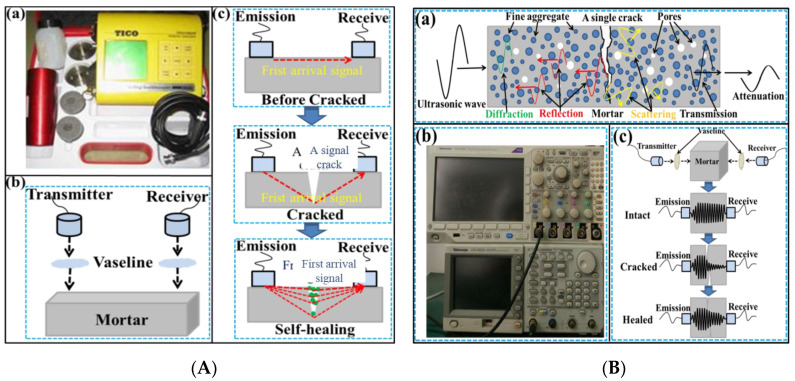
UPV (**A**) and transmitted ultrasonic wave (**B**) test.

**Figure 5 materials-16-03453-f005:**
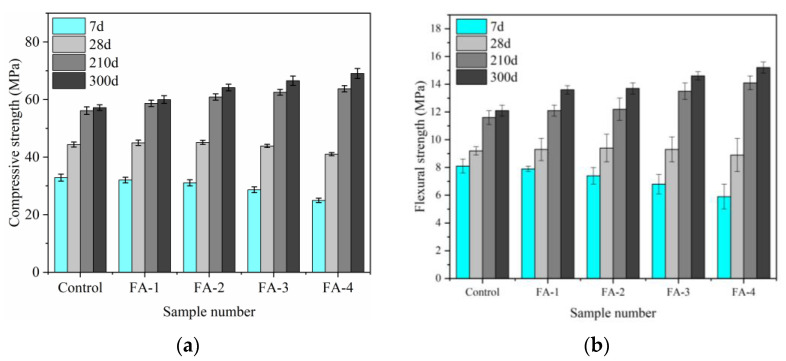
Effect of fly ash on mortar mechanical strength. (**a**) compressive strength. (**b**) flexural strength.

**Figure 6 materials-16-03453-f006:**
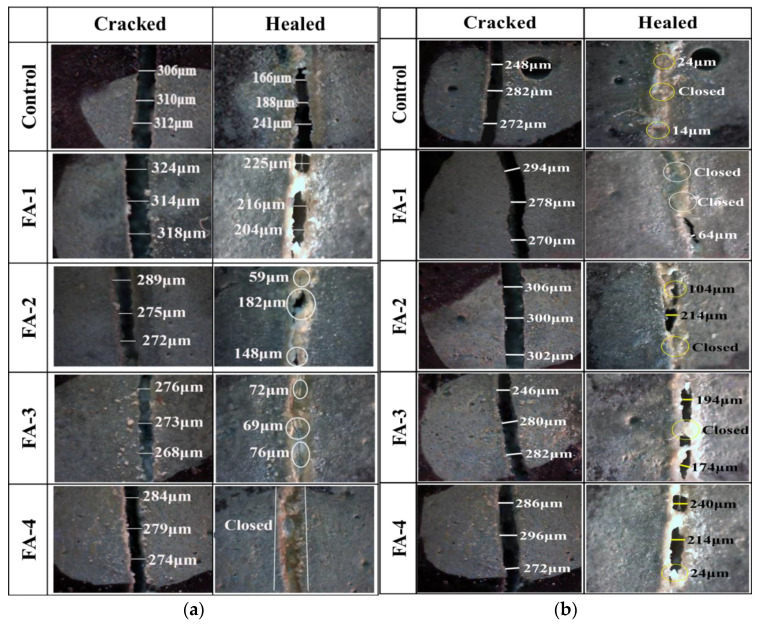
Influence of fly ash content on cracks closure of samples cured 28 d (**a**) and 210 d (**b**).

**Figure 7 materials-16-03453-f007:**
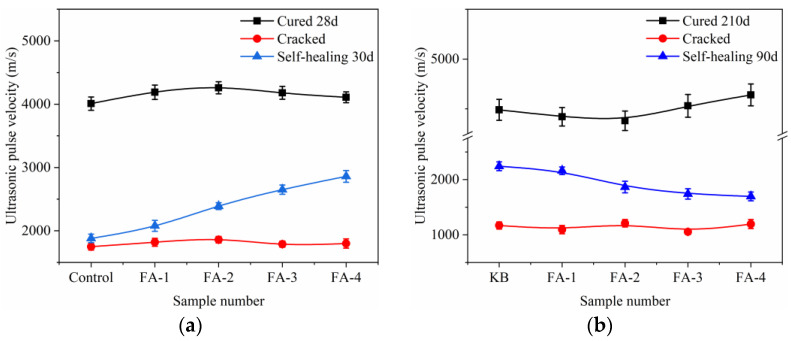
Influence of FA on the UPV cured 28 d (**a**) and 210 d (**b**).

**Figure 8 materials-16-03453-f008:**
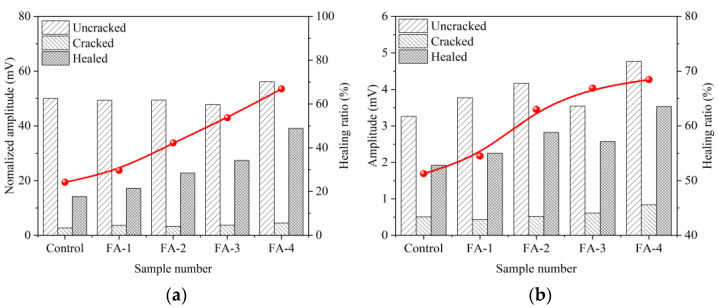
Effect of fly ash content on healing ratio of maximum amplitude (**a**) and frequency amplitude (**b**) at 28 d.

**Figure 9 materials-16-03453-f009:**
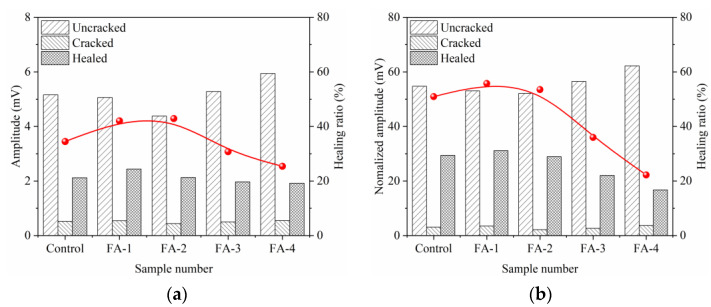
Effect of fly ash dosage on healing ratio of wave amplitude (**a**) and frequency amplitude (**b**) at 210 d.

**Figure 10 materials-16-03453-f010:**
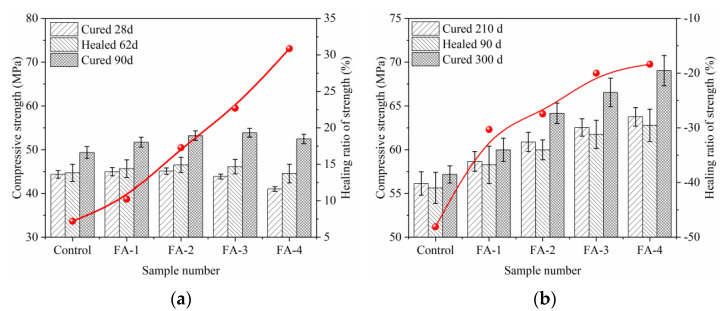
Effect of fly ash on strength recovery of cracked samples cured 28 d (**a**) and 210 d (**b**).

**Figure 11 materials-16-03453-f011:**
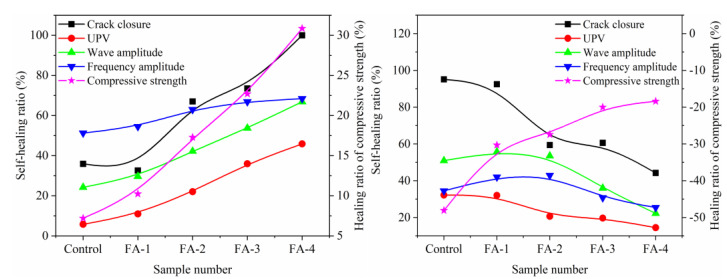
Correlation analysis of FA content and self-healing ratios.

**Figure 12 materials-16-03453-f012:**
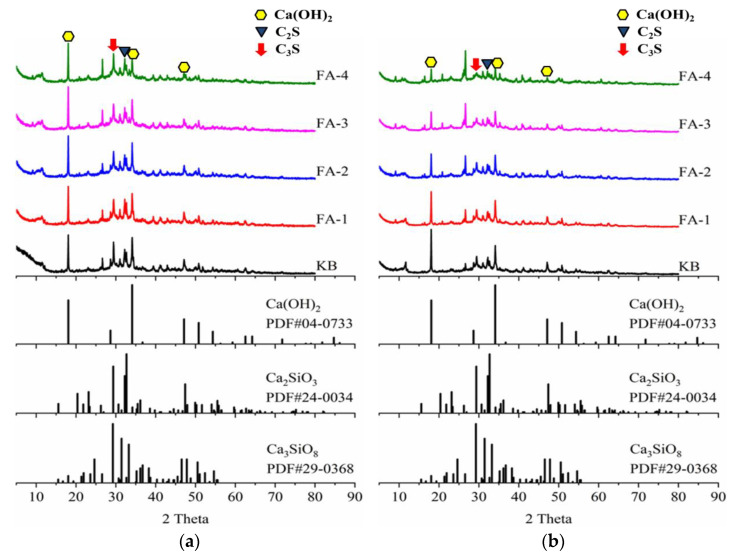
XRD analysis of hydration products for samples cured 28 d (**a**) and 210 d (**b**).

**Figure 13 materials-16-03453-f013:**
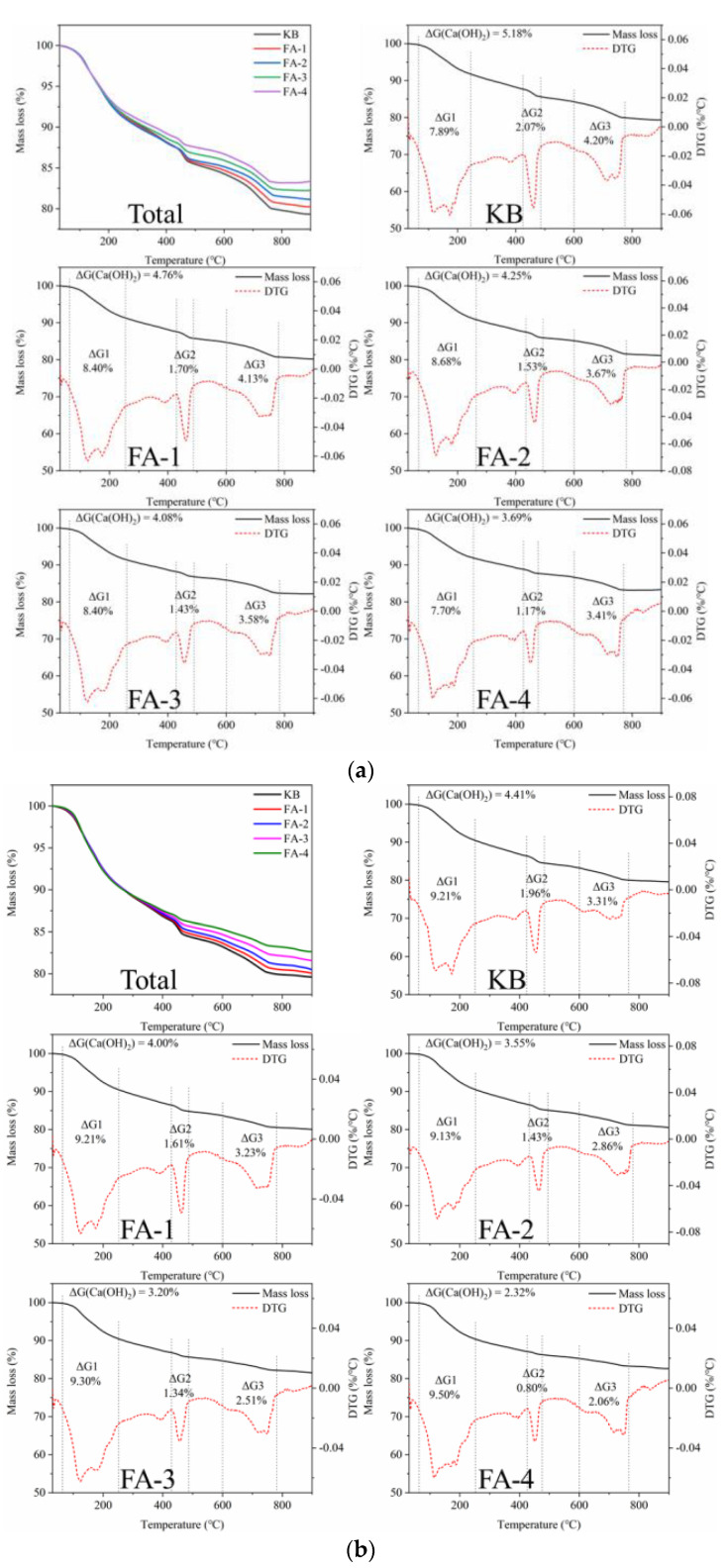
TG analysis of hydration products for samples cured for 28 d (**a**) and 210 d (**b**).

**Figure 14 materials-16-03453-f014:**
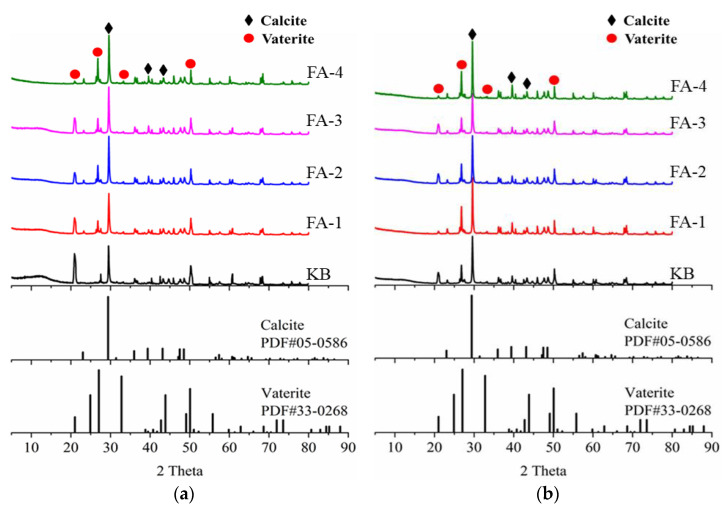
XRD analysis of self-healing products at cracks produced in samples cured for 28 d (**a**) and 210 d (**b**).

**Figure 15 materials-16-03453-f015:**
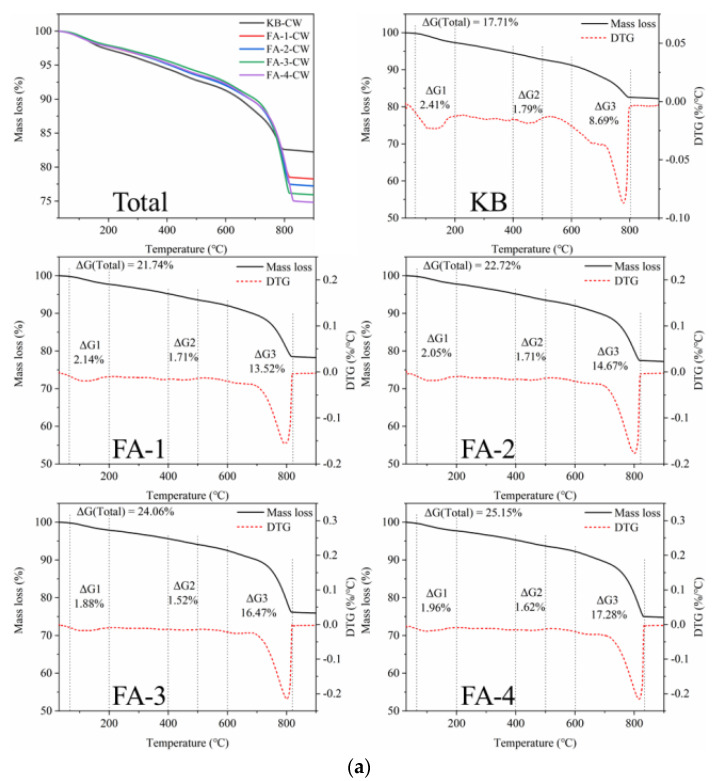
TG analysis of self-healing products at cracks produced in samples cured for 28 d (**a**) and 210 d (**b**).

**Figure 16 materials-16-03453-f016:**
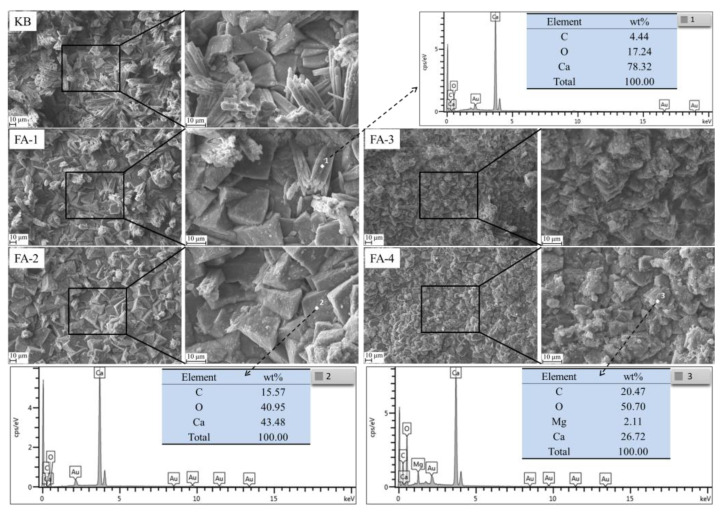
SEM-EDS analysis of self-healing products in cracks produced in samples cured for 28 d.

**Figure 17 materials-16-03453-f017:**
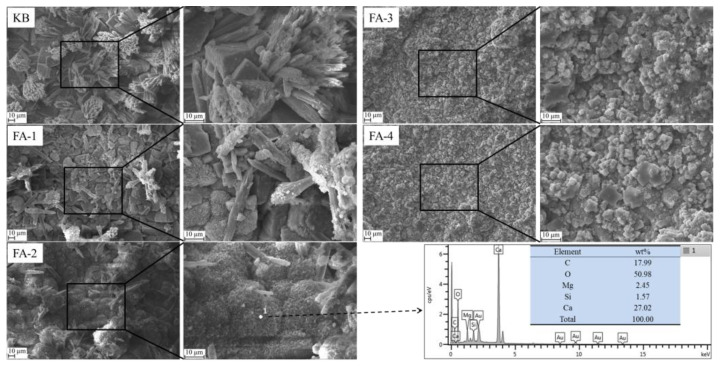
SEM-EDS analysis of self-healing products in cracks produced in samples cured for 210 d.

**Table 1 materials-16-03453-t001:** Chemical compositions of P·O 42.5 and FA (%).

Compositions	CaO	SiO_2_	Al_2_O_3_	MgO	Fe_2_O_3_	SO_3_	Others	Specific Surface Area (m²/kg)
P·O 42.5	57.82	22.16	6.36	2.94	2.78	4.20	3.74	346
fly ash	3.57	48.59	37.08	0.36	5.24	0.34	4.82	445

**Table 2 materials-16-03453-t002:** Particle size distribution of river sand (%).

Size Distribution (mm)	≤0.15	0.15~0.3	0.3~0.6	0.6~1.18	1.18~2.36	2.36~4.75
Content (%)	11	21	27	24	12	5

**Table 3 materials-16-03453-t003:** Mix proportion of mortar.

No.	Cement/g	Fly Ash/g	Water/g	River Sand/g
Control	450	0	150	1350
FA-1	405	45		
FA-2	360	90		
FA-3	315	135		
FA-4	270	180		

**Table 4 materials-16-03453-t004:** Required conditions for different mechanisms of self-healing.

Mechanisms	Conditions and Results			
	Presence of water in cracks	Presence of ${\mathrm{CO}}_{3}^{2-}$ ions in cracks	Upper crack width that can be repaired	Compressive strength
Autogenous self-healing	In a watery environment	Not necessary. However, the presence of ${\mathrm{CO}}_{3}^{2-}$ is better.	Less than 50 μm	Compressive strength loss decreases from 27% to 7% [14,34]
Self-healing based on mineral admixtures	In a watery environment	Not necessary.	About 200 μm	Recovering 85%/(74%) of initial resonant frequency [35]
Self-healing based on bacteria	In a watery environment	${\mathrm{CO}}_{3}^{2-}$ion is needed	Less than 450 μm	A slight increase in 28 d of curing. [36]
Self-healing based on adhesive agents	In most cases, cracks should be in a watery environment	Not necessary.	Depending on the amount of agents released	Depending on healing agent [37].

## Data Availability

Not applicable.

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
