# Peer review of "Uncovering the Mechanism of the Role of Fly Ash in the Self-Healing Ability of Mortar with Different Curing Ages"

_materials, 2023, doi:10.3390/ma16093453_

Round 1

Reviewer 1 Report

1. The work does not respect the Template. Check the references.

2. The author should refine all chapter of the paper.

3. Lines 73-74: “In the paper, the P.O 42.5(ordinary Portland cement? please use the correct name. In 73

the EN standards it is ordinary Portland cement CEM I 42,5 N. Refer to your standard.)” - ??? I don't understand

4. Describe the methodology for determining the chemical composition of P.O 42.5 and FA (%).

5. How it was determined Fe2O3?

6. How it was determined SO3?

7. Describe the methodology for determining the particle size distribution of river sand.

8. Justify how the proportions were chosen the “Mix proportion of mortar”- Table 3.

9. The resolution of Figure 13 is poor.

10. The resolution of Figure 15 is poor.

11. Insert a flow chart for your research.

12. What is the novelty of the paper?

13. What is the added value of the paper?

14. A comparative analysis between the results obtained and those reported in the specialized literature is necessary. Insert a Table. A scientific discussion is necessary.

15. “Figure 16. SEM-EDS analysis of self-healing products at cracks produced in samples cured 28 d.” - In EDS what does "Au" represent.

Author Response

Review 1

Responses: Thank you very much for your kind letter and the reviewers’ comments. Those comments are very useful for improving our research and revising our manuscript. Based on the suggestions and comments, we have made relevant amendments and we hope the paper is more readable.

  1. The work does not respect the Template. Check the references.

Responses: All literature formats have been modified

2.The author should refine all chapter of the paper.

Responses: Thank you for your suggestions. The language of the article has been enhanced

  1. Lines 73-74: “In the paper, the P.O 42.5(ordinary Portland cement? please use the correct name. In 73

the EN standards it is ordinary Portland cement CEM I 42,5 N. Refer to your standard.)” - ??? I don't understand

Responses: In the paper, Portland cement (P·O 52.5) was bought from Shanshui Co., Ltd. the P.O 42.5 and FA was supplied by the Shandong Lubi Co., Ltd.

  1. Describe the methodology for determining the chemical composition of P.O 42.5 and FA (%).

Responses: The chemical composition of FA and cement were analyzed by X-ray fluorescence spectrometer (S8 TIGER, Brook) is recorded in Table 1. The principle is based on the X-ray intensity of different elements, to determine the content of elements

  1. How it was determined Fe2O3?

Responses: According to the X-ray intensity of iron, to determine the content of iron element, calculate the content of Fe2O3.

  1. How it was determined SO3?

Responses: According to the X-ray intensity of sulfur, to determine the content of sulfur element, calculate the content of SO3.

  1. Describe the methodology for determining the particle size distribution of river sand.

Responses: Put the sieves of different particle sizes, from small to large in order. Put 500g sieve into the largest aperture sieve, put it on the vibrating sieve machine after vibration, calculate the percentage of sieve on each level of sieve. The results was listed in Table 2.

  1. Justify how the proportions were chosen the “Mix proportion of mortar”- Table 3.

Responses: The water-binder ratio of common concrete is generally about 0.4-0.45, which meets the design standard of C40. In this paper, a more conventional design standard is chosen to meet most engineering standards.

Table 3

Mix proportion of mortar

No.

Cement /g

Fly ash /g

Water /g

River sand /g

Control

450

0

150

1350

FA-1

405

45

FA-2

360

90

FA-3

315

135

FA-4

270

180

  1. The resolution of Figure 13 is poor.

(a)

(b)

Fig. 13 TG analysis of hydration products for samples cured 28 d (a) and 210 d (b)

  1. The resolution of Figure 15 is poor.

(a)

(b)

Fig. 15 TG analysis of self-healing products at cracks produced in samples cured 28 d (a) and 210 d (b)

  1. Insert a flow chart for your research.

Responses: A flow chart has been added.

  1. What is the novelty of the paper?

Responses: As an admixture of cement-based materials, the reaction of fly ash (FA) usually takes place in the late age, so FA will affect the self-healing ability of long-age cement-based materials. The main innovation of this paper is to explore the influence mechanism of fly ash on the self-healing ability of cement after 210 days of curing.

  1. What is the added value of the paper?

Responses: The main value of this paper is to improve the theoretical basis for intrinsic self-healing and durability improvement of cement based materials in practice

  1. A comparative analysis between the results obtained and those reported in the specialized literature is necessary. Insert a Table. A scientific discussion is necessary.

Responses:

Table 4 provides a summary and comparison of the cracking and strength recovery achieved through different self-healing mechanisms. In this study, the use of 40% fly ash in mortar resulted in cross-strength crack healing and strength recovery after 30 days. The fly ash was able to effectively heal cracks up to 270 micrometers with good strength recovery, and was found to be comparable in performance to self-healing concrete incorporating mineral admixtures or bacteria. Overall, the results suggest that the use of fly ash has significant potential for improving the self-healing capabilities of concrete.

Table 4

Required conditions for different mechanisms of self-healing.

Mechanisms

Conditions and results

Presence of water in cracks

Presence of  ions in cracks

Upper crack width that can be repaired

Compressive strength

Autogenous

self-healing

In a watery environment

Not necessary. But the presence of  is better.

Less than

50 μm

Compressive strength loss decreases from 27% to 7%[14, 31]

Self-healing based

on mineral

admixtures

In a watery environment

Not necessary.

About 200 μm

Recovering 85%/ (74%) of initial resonant frequency[32]

Self-healing based

on bacteria

In a watery environment

ion is needed

Less than 450 μm

A slight increase in 28 days cured.[33]

Self-healing based on adhesive agents

In most cases cracks should be in a watery environment

Not necessary.

Depending on the amount of agents released

Depending on healing agent [34].

  1. “Figure 16. SEM-EDS analysis of self-healing products at cracks produced in samples cured 28 d.” - In EDS what does "Au" represent.

Responses: "Au" represents the gold. Because the concrete is non-conductive, it is necessary to spray a layer of gold powder on the surface of the sample to increase the electrical conductivity of the sample. In this way, the SEM-EDS pictures of the sample can be better observed.

Reviewer 2 Report

The article “Investigation self-healing ability of long-age curing mortar with different fly ash content” deals with an interesting topic using a wide spectrum of test results. However, the following major aspects should be addressed before acceptance:

- Improve the conclusions and results. After presenting UPV tests, XRD, TG, SEM-EDS, OM measurements, the three conclusions are quite expected. For example, include future research for the problems identified with autogeneous self-healing using FA. Also, consider to enhance the comparison with results presented by other researchers worldwide.

- Strenghen the methodology. There is a lack of important information in some tests. For example, brand, model, sensibility, calibration from the equipment used for the UPV tests will help to compare with similar results. This is information is also important for XRD, TG, SEM-EDS, OM. Furthermore, transfer some information from results, e.g. equations 1 to 3, to methodology. Finally, include a column on Table 3 with the FA percentages compare to the binder phase.

- Enhance the figures description. In most cases, there is only a general title which does not explain a sequence of pictures with letters. For example, in Figure 4 letters a to c are not explained. Verify this ascpect for the 17 figures included in the article.

- Before submitting, check carefully to complete the internal review process with your co-authors. It seems some sentences are still in the manuscript from your co-authors. For example, line 73 “please use the correct name…” and 147 “what do you mean on late strength?”. Check carefully the article grammar. In my opinión, a preposition “on” should be added in the title between “Investigation” and “self-healing”. Similar issues along the article.

Author Response

The article “Investigation self-healing ability of long-age curing mortar with different fly ash content” deals with an interesting topic using a wide spectrum of test results. However, the following major aspects should be addressed before acceptance:

Responses: Thank you very much for your approval and comments on our work. The paper has been carefully revised according to your valuable suggestions. Relevant amendments have been made as follows:

- Improve the conclusions and results. After presenting UPV tests, XRD, TG, SEM-EDS, OM measurements, the three conclusions are quite expected. For example, include future research for the problems identified with autogeneous self-healing using FA. Also, consider to enhance the comparison with results presented by other researchers worldwide.

Responses:

  1. The increase of fly ash content promoted the improvement of self-repair performance of cracked specimens at 28 d, especially when the fly ash dosage was 40%, the crack opening after 30 days of healing in water was completely closed, the UPV value after recovery was close to 3000 m/s, the self-repair efficiency of maximum amplitude and main frequency amplitude was up to more than 60%, and the recovery rate of compressive strength was increased to know more than 30%. However, the increase of fly ash content was not conducive to the self-repair of cracked samples at 210 d, and with the increase of fly ash content, the crack closure effect weakened, the UPV value after recovery decreased, and the crack repair rate based on ultrasonic transmission decreased to about 20%, and the compressive strength recovery rate increased slightly. The pozzolanic reaction of fly ash improved the self-repair performance of cementitious materials, especially the strength recovery; however, the absence of Ca2+ in fly ash inhibited the precipitation of calcium carbonate, which was not conducive to crack filling.
  2. The initial crack surface of the 28 d samples contained calcium hydroxide and unreacted cement clinker minerals, which provided great potential for crack self-repair; whereas the crack surface of the 210 d samples contained less calcium hydroxide and unhydrated cement clinker minerals, and the intensity of the diffraction peak of calcium hydroxide gradually decreased with the increase of fly ash admixture and the content decreased.
  3. Calcium carbonate precipitation was the main repair product of crack filling and healing, including calcite and spherulite. With the decrease of fly ash admixture, the content of calcium hydroxide increased, which promoted the precipitation of spherulite. With the reaction of fly ash, the precipitation of calcite increased. With the increase of fly ash content, the content of element C in the self-repair products of 28-day-old specimens gradually increased, and the size of calcium carbonate crystals gradually decreased, but the filling was more dense; while the calcium carbonate crystals in the self-repair products of 210d specimens gradually became fine and loose.

Table 4 provides a summary and comparison of the cracking and strength recovery achieved through different self-healing mechanisms. In this study, the use of 40% fly ash in mortar resulted in cross-strength crack healing and strength recovery after 30 days. The fly ash was able to effectively heal cracks up to 270 micrometers with good strength recovery, and was found to be comparable in performance to self-healing concrete incorporating mineral admixtures or bacteria. Overall, the results suggest that the use of fly ash has significant potential for improving the self-healing capabilities of concrete.

Table 4

Required conditions for different mechanisms of self-healing.

Mechanisms

Conditions and results

Presence of water in cracks

Presence of  ions in cracks

Upper crack width that can be repaired

Compressive strength

Autogenous

self-healing

In a watery environment

Not necessary. But the presence of  is better.

Less than

50 μm

Compressive strength loss decreases from 27% to 7%[14, 31]

Self-healing based

on mineral

admixtures

In a watery environment

Not necessary.

About 200 μm

Recovering 85%/ (74%) of initial resonant frequency[32]

Self-healing based

on bacteria

In a watery environment

ion is needed

Less than 450 μm

A slight increase in 28 days cured.[33]

Self-healing based on adhesive agents

In most cases cracks should be in a watery environment

Not necessary.

Depending on the amount of agents released

Depending on healing agent [34].

- Strenghen the methodology. There is a lack of important information in some tests. For example, brand, model, sensibility, calibration from the equipment used for the UPV tests will help to compare with similar results. This is information is also important for XRD, TG, SEM-EDS, OM. Furthermore, transfer some information from results, e.g. equations 1 to 3, to methodology. Finally, include a column on Table 3 with the FA percentages compare to the binder phase.

Responses: The missing information has been added.

2.3.4 Mechanical strength

On the 7 d, 28 d, 210 d and 300 d, the flexural and compressive strength was tested according to GB/T17617-1999 (Chinese standard).

Ultrasonic pulse velocity (UPV) test has been widely used in concrete structure detection, and reliable results have been obtained in the characterization of crack healing. UPV and transmitted ultrasonic wave test instruments and their mechanism are shown in Fig. 4. Before the test, the mortar was dried at 50 â—¦C for 24 h, eliminating the absorption of ultrasonic energy by free water. The device was provided by Shenborui Company, China. The natural frequency of sensors was 54 kHz, the pulse voltage was 1 kV, the resolution was 0.1 μs. The transmitted ultrasonic frequency was 150kHz, which was sent out by the pulse transmitter (Tektronix AFG 3022C) and received by the mixed domain oscilloscope (Tektronix MDO 3024). The loss rate was calculated to determine the self-healing rate. In addition, the frequency data was obtained by the fast Fourier transform.

The hydration products of paste and self-healing were collected, dried, and broken into powder for XRD (Brook D8 Advance, Germany) and TG tests. The powder was heated from 25 â—¦C to 1000 â—¦C at 10 â—¦C/min in argon atmosphere for TG and scanned from 5â—¦ to 80â—¦ with a step size of 0.02â—¦ for XRD analysis. The specimen was dried at 60 â—¦C for 24h and then covered by gold to wait for the detection. The microscopic morphology of self-healing products with different curing ages was observed by the SEM-EDS (ZeissEVO LS 15, Germany).

Table 3

Mix proportion of mortar

No.

Cement /g

Fly ash /g

Water /g

River sand /g

Control

450

0

150

1350

FA-1

405

45

FA-2

360

90

FA-3

315

135

FA-4

270

180

- Enhance the figures description. In most cases, there is only a general title which does not explain a sequence of pictures with letters. For example, in Figure 4 letters a to c are not explained. Verify this ascpect for the 17 figures included in the article.

Responses: The figures description in this paper have been enhanced, such as Figure 4.

Ultrasonic pulse velocity (UPV) test has been widely used in concrete structure detection, and reliable results have been obtained in the characterization of crack healing. UPV and transmitted ultrasonic wave test instruments and their mechanism are shown in Fig. 4. Before the test, the mortar was dried at 50 â—¦C for 24 h, eliminating the absorption of ultrasonic energy by free water. The device was provided by Shenborui Company, China. The natural frequency of sensors was 54 kHz, the pulse voltage was 1 kV, the resolution was 0.1 μs. The transmitted ultrasonic frequency was 150kHz, which was sent out by the pulse transmitter (Tektronix AFG 3022C) and received by the mixed domain oscilloscope (Tektronix MDO 3024). Besides, some measures were conducted to decrease the influence of external factor: First, all ultrasonic test was completed in interior environment. Second, the surface contacted with transducer was polished by 1000 # abrasive paper. The thickness loss could be neglected. Vaseline was applied on the interface evenly to enhance the coupling of transducer and specimen. Third, transducer was at the same the position in each ultrasonic test. The loss rate was calculated to determine the self-healing rate. In addition, the frequency data was obtained by the fast Fourier transform. In PUV test, the side face of transducer and specimen was aligned. The distance of two sensors was determined as 110 mm. In ultrasonic wave test, the sensors were at the two ends of specimen. The distance was 160 mm. Although the amplitude value would slightly change in repeated test of one specimen, the amplitude tendency of uncrack, cracked and repaired specimen is regular and repeatable.

- Before submitting, check carefully to complete the internal review process with your co-authors. It seems some sentences are still in the manuscript from your co-authors. For example, line 73 “please use the correct name…” and 147 “what do you mean on late strength?”. Check carefully the article grammar. In my opinión, a preposition “on” should be added in the title between “Investigation” and “self-healing”. Similar issues along the article.

Responses: Thank you for your comments. The whole article has been modified under your suggestion. Excised the superfluous part and fixed the grammatical errors in the article. It has modified the problem of excessive improper prepositions to improve the quality and readability of the article.

Reviewer 3 Report

Although the manuscript is relevant to the journal's topics, it cannot currently be recommended for publication.
The language is fairly basic, hence a native expert proofreader is needed.

The text of the manuscript is rather sloppy, and certain comments phrases have been left uncorrected. This demonstrates a lack of commitment prior to submission.
To generalize the findings, figures 13 and 14's quality should be increased, and the conclusion should be restated in a more inclusive manner.

For all these reasons the reviewer suggests a new submission and a new review process after taking into account all the previous comments

Author Response

Although the manuscript is relevant to the journal's topics, it cannot currently be recommended for publication.

Responses: Thank you very much for your kind letter and the reviewers’ comments. Those comments are very useful for improving our research and revising our manuscript.

The language is fairly basic, hence a native expert proofreader is needed.

Responses: Based on your comments, we have made relevant amendments and we hope the paper is more readable.

The text of the manuscript is rather sloppy, and certain comments phrases have been left uncorrected. This demonstrates a lack of commitment prior to submission.

Responses: Thanks for your comments, the quality of manuscripts was improved and some certain comments phrases have been removed.

To generalize the findings, figures 13 and 15's quality should be increased, and the conclusion should be restated in a more inclusive manner.

Responses: The corresponding image has been modified.

(a)

(b)

Fig. 13 TG analysis of hydration products for samples cured 28 d (a) and 210 d (b)

(a)

(b)

Fig. 15 TG analysis of self-healing products at cracks produced in samples cured 28 d (a) and 210 d (b)

For all these reasons the reviewer suggests a new submission and a new review process after taking into account all the previous comments

Responses: Thank you again, the revised manuscript has been submitted

Reviewer 4 Report

The manuscript tackles a very interesting topic, the experiments are oriented towards the goal of the paper and every observation is very well documented and explained.

There are a few issues that need to be addressed before the manuscript could be accepted for publication.

In the Introduction section the authors talk mainly about concrete and only towards the end mortar is mentioned. I think part of this section should include more information related to mortar than concrete.

Line 47 - please give the full name before using acronyms (ECC).

Line 53 - please substitute "pointed" by "showed" or "demonstrated"

Line 59 - "FA promoted ..."

Line 69 - please substitute "calculated" by "assessed" / "determined"

Lines 15 and 17 - the sentences seem unfinished "ultrasonic....?"

Line 23 - cracks can not create durability but can create durability issues or problems.

Line 39 - please pay attention to the units of measure. Shouldn't it be nm instead of mm?

Lines 73-74 - please remove the unnecessary comment

Line 75 - "the chemical composition of which was determined"

Line 79 - "fixed at..."

Line 80 - please substitute "mixed' by "mix"

Figures 1 and 2 - please indicate the identified components on the XRD spectra

Line 122 - "frequency data"

Figure 4 - please distinguish between UPV and ultrasonic wave test more clearly. In UPV diagram c please correct "frist" to "first"

Lines 138-140 - Please rephrase this sentence. I think the authors wanted to say that by using FA to replace OPC, the quantity of OPC decreased resulting in a decreased amount of hydration products. Also, please substitute "past" by "paste"

Lines 143-144 - please rephrase

Line 144 - "after curing for 210d..."

Lines 147-150 - please remove the unnecessary comment

Figures 8a and 9b - vertical axis - Normalized

Line 368 - please replace "bully" by "fully"

Line 376 - "rod-shaped"

As a general comment - please use present tense when you refer to figures and tables. For example "is presented if Fig. / is shown if Fig."

Author Response

Review 4

The manuscript tackles a very interesting topic, the experiments are oriented towards the goal of the paper and every observation is very well documented and explained. There are a few issues that need to be addressed before the manuscript could be accepted for publication.

Responses: Thank you very much for your kind letter and the reviewers’ comments. Those comments are very useful for improving our research and revising our manuscript. Based on the suggestions and comments, we have made relevant amendments and we hope the paper is more readable.

In the Introduction section the authors talk mainly about concrete and only towards the end mortar is mentioned. I think part of this section should include more information related to mortar than concrete.

Responses: The self-healing properties of the mortar have been added.

Line 47 - please give the full name before using acronyms (ECC).

Responses: Mustafa Sahmaran et al[15] have found engineered cementitious composite(ECC) samples with FA had a higher self-healing ability than those with slag.

Line 53 - please substitute "pointed" by "showed" or "demonstrated"

Responses: The "pointed" has been replaced by "demonstrated".

They demonstrated the self-healing deflection capacity of pre-cracked samples at 14 and 28 days recovered or exceeded that of the original samples at almost all pre-cracked ages.

Line 59 - "FA promoted ..."

Responses: The fly ash has been added before the acronyms of FA.

Line 69 - please substitute "calculated" by "assessed" / "determined"

Responses: The "calculated" has been replaced by "determined".

Lines 15 and 17 - the sentences seem unfinished "ultrasonic....?"

Responses: The corresponding sentence has been modified.

Line 23 - cracks can not create durability but can create durability issues or problems.

Responses: The Crack is one of the main factors causing concrete damage and durability issues

Line 39 - please pay attention to the units of measure. Shouldn't it be nm instead of mm?

Responses: The mistake has been revised.

Lines 73-74 - please remove the unnecessary comment

Responses: The unnecessary comment has been removed.

Line 75 - "the chemical composition of which was determined"

Responses: The chemical composition of FA and cement was recorded in Table 1.

Line 79 - "fixed at..."

Responses: The water to binding materials ratio and the binding materials to river sand ratio of mortar were fixed at 0.4 and 1:3, respectively.

Line 80 - please substitute "mixed' by "mix"

Responses: The mix proportion of mortar was shown in Table 3.

Figures 1 and 2 - please indicate the identified components on the XRD spectra

Responses: The corresponding phase components have been added.

Fig. 2 XRD pattern of fly ash and P·O 42.5

Line 122 - "frequency data"

Responses: In addition, the frequency data was obtained by the FFT.

Figure 4 - please distinguish between UPV and ultrasonic wave test more clearly. In UPV diagram c please correct "frist" to "first"

Responses: The transmitted ultrasonic frequency was 150kHz, which was sent out by the pulse transmitter (Tektronix AFG 3022C) and received by the mixed domain oscilloscope (Tektronix MDO 3024). The loss rate was calculated to determine the self-healing rate.

The "frist" has been replaced with the "first".

Fig. 4 UPV and transmitted ultrasonic wave test

Lines 138-140 - Please rephrase this sentence. I think the authors wanted to say that by using FA to replace OPC, the quantity of OPC decreased resulting in a decreased amount of hydration products. Also, please substitute "past" by "paste"

Responses: The mistake has been revised.

Because the fly ash replaced cement, the quality of cement decreased resulting in a decrease amount of hydration products. In addition, the low activity of fly ash in early stage produced little hydration products and cannot meet its dilution effect, so the early strength of mortar decreased.

Lines 143-144 - please rephrase

Responses: With the curing age increasing, FA reacted with CH and produced new hydration products, which filled the pores and the microstructure and improved mechanical properties of mortar.

Line 144 - "after curing for 210d..."

Responses: The "after curing 210d..." has been replaced with the "After curing for 210d...".

Lines 147-150 - please remove the unnecessary comment

Responses: The unnecessary comment has been removed

Figures 8a and 9b - vertical axis – Normalized

Responses: Thank you for your suggestion. We tried to unify the vertical axis before, but the drawing was not beautiful and the result could not be seen directly. This is the result of several adjustments and is the final picture.

Line 368 - please replace "bully" by "fully"

Responses: The "bully " has been replaced with the " fully ".

Line 376 - "rod-shaped"

Responses: The " rode-shaped " has been replaced with the " rod-shaped ".

As a general comment - please use present tense when you refer to figures and tables. For example "is presented if Fig. / is shown if Fig."

Responses: Thank you for your suggestion. When refer the tables or graphs, the present tense is usually used.

Reviewer 5 Report

Before the article is accepted for publication, it is important to answer the following information:

- The title of the article does not make it clear what the main research points are. Please review this information.

- In the abstract include the main objectives of the article. Make it clear which gaps are being evaluated. Include the main results and main conclusions obtained in the research.

- The introduction approach is flawed. After reading this section, it is important to make it clear to readers what are the main points being investigated. That is why it is important to highlight the information that is already known about tem (self-healing ability of long-age and influence of fly ash in mortars) so that the originality of the research becomes clear. There are many works that evaluate the influence of fly ash in mortars. Why is your study different? Why does it deserve to be published? This is not made clear in the introduction.

- The granulometry of the materials must be included in a single figure. I believe that all raw materials used must have the granulometry included in the form of a granulometric curve, preferably in a single image. It helps to compare the materials used.

- In Figure 1 and 2, identify the peaks present in the XRD. A figure with this pattern has no validity in the paper if the crystalline peaks are not identified. Please fix this.

- Is the composition used 1:3:0.4 (cement: sand: water)? If so, use the accepted standard for presenting mortar compositions. Explain how this composition was defined. What are your applications?

- Standardize Figure 5. Why use a figure in columns and another in lines? Does not make sense. Review this point throughout the article.

- Usually at 28 days, the Portland cement hydration process already reaches high levels. In his research, the resistance gain between 28 and 210 days is intensified, contrary to some research. Why does it happen?

- Figure 5: this effect of FA on mortars is widely known. What's new in research? All authors in the field of construction materials are aware of this behavior. I have many doubts about the originality of the research because the authors do not discuss anything new. Compare the results with similar research and explain what gaps are being studied.

- See the formatting of equations (1), (2) and (4)

- Figure 12: the authors evaluate the Portland cement hydration process with FA but in the XRD identification, they observe the presence of C2S and C3S even after 28 and 210 days of hydration. Is this correct? C2S and C3S are not hydration products. They are present in anhydrous cement. This is incorrect.

- Review conclusions after responding to previous comments.

Author Response

Before the article is accepted for publication, it is important to answer the following information:

Responses: Thank you very much for your kind letter and the reviewers’ comments. Those comments are very useful for improving our research and revising our manuscript. Based on the suggestions and comments, we have made relevant amendments and we hope the paper is more readable.

- The title of the article does not make it clear what the main research points are. Please review this information.

Responses: Uncovering the mechanism of the role of fly ash in the self-healing ability of mortar with different curing ages.

- In the abstract include the main objectives of the article. Make it clear which gaps are being evaluated. Include the main results and main conclusions obtained in the research.

Responses: As an admixture of cement-based materials, the reaction of fly ash (FA) usually takes place in the late age, so FA will affect the self-healing ability of long-age cement-based materials. The self-healing potential and the characteristics of self-healing products of cementitious materials before and after crack healing were analyzed by microscopic tests, and the mechanism of the effect of fly ash on the self-healing performance of cementitious materials was revealed. Results showed that the increase of fly ash content promoted the improvement of self-repair performance of cracked specimens at 28 d, especially when the fly ash dosage was 40%, the crack opening after 30 days of healing in water was completely closed, the UPV value after recovery was close to 3000 m/s, the self-repair efficiency of maximum amplitude and main frequency amplitude was up to more than 60%, and the recovery rate of compressive strength was increased to know more than 30%. However, the increase of fly ash content was not conducive to the self-repair of cracked samples at 210 d, and with the increase of fly ash content, the crack closure effect weakened, the UPV value after recovery decreased, and the crack repair rate based on ultrasonic transmission decreased to about 20%, and the compressive strength recovery rate increased slightly. In addition, calcium carbonate precipitation was the main repair product of crack filling and healing, including calcite and spherulite. With the increase of fly ash content, the content of element C in the self-repair products of 28-day-old specimens gradually increased, and the size of calcium carbonate crystals gradually decreased, but the filling was more dense; while the calcium carbonate crystals in the self-repair products of 210d specimens gradually became fine and loose.

- The introduction approach is flawed. After reading this section, it is important to make it clear to readers what are the main points being investigated. That is why it is important to highlight the information that is already known about tem (self-healing ability of long-age and influence of fly ash in mortars) so that the originality of the research becomes clear. There are many works that evaluate the influence of fly ash in mortars. Why is your study different? Why does it deserve to be published? This is not made clear in the introduction.

Responses: The corresponding changes are added.

As a common cementitious material, fly ash played a significant role in workability and long-age performance development in concrete. Due to its pozzolanic reaction, fly ash underwent secondary hydration, consumed CH, and formed new C-S-H gel in the later stage, which changed the microstructure and impacted the self-healing ability of concrete. Pipat Termkhajornkit et al.[17] have demonstrated that the fly ash–cement system has the self-healing ability for cracks that occur from shrinkage, and the self-healing ability increased when the fraction of fly ash increased. But The research focuses on behavior after 28 days. Zhigang Zhang et al. [18] have pointed that the secondary hydration of fly ash in the later stage can change the micro-structure of concrete and also show positive impact on the self-healing behavior. While self-healing with FA in concrete is available, the self-healing ability of mortar with FA after long-term curing is limited in the literature.

- The granulometry of the materials must be included in a single figure. I believe that all raw materials used must have the granulometry included in the form of a granulometric curve, preferably in a single image. It helps to compare the materials used.

Fig. 1 Particle size of P·O 42.5 and fly ash

- In Figure 1 and 2, identify the peaks present in the XRD. A figure with this pattern has no validity in the paper if the crystalline peaks are not identified. Please fix this.

Responses: Corresponding phase components are added.

- Is the composition used 1:3:0.4 (cement: sand: water)? If so, use the accepted standard for presenting mortar compositions. Explain how this composition was defined. What are your application ns?

Responses: General grade C30, C35 concrete ratio of water-cement ratio of about 0.4, this grade of concrete is the pile foundation of road projects, housing projects, the foundation of the commonly used concrete grade.

- Standardize Figure 5. Why use a figure in columns and another in lines? Does not make sense. Review this point throughout the article.

Responses: The corresponding picture is modified.

(a)                                   (b)

Fig. 5 Effect of FA on mortar mechanical strength

- Usually at 28 days, the Portland cement hydration process already reaches high levels. In his research, the resistance gain between 28 and 210 days is intensified, contrary to some research. Why does it happen?

Responses: Before 28d, a very small amount of fly ash participated in the reaction. With the increase of fly ash content, the amount of cement was reduced, the amount of hydration products was reduced, and the compactness and mechanical properties of mortar were reduced. When the mortar was cured to 210 d, most of the fly ash in the hydration reaction, producing new hydration products and increasing the compactness of motar. With the increase of fly ash content, the strength has the opposite result.

- Figure 5: this effect of FA on mortars is widely known. What's new in research? All authors in the field of construction materials are aware of this behavior. I have many doubts about the originality of the research because the authors do not discuss anything new. Compare the results with similar research and explain what gaps are being studied.

Responses: The main purpose of Fig. 5 is not to study the influence of fly ash on mortar performance, but to lay a foundation for subsequent verification test results, that is, to study the strength change of mortar after self-healing repair. The data in this figure only serves as a reference.

- See the formatting of equations (1), (2) and (4)

Responses: The corresponding formula is edited with the mathematical formula editor.

- Figure 12: the authors evaluate the Portland cement hydration process with FA but in the XRD identification, they observe the presence of C2S and C3S even after 28 and 210 days of hydration. Is this correct? C2S and C3S are not hydration products. They are present in anhydrous cement. This is incorrect.

Responses: We fully agree with you that C2S and C3S are not a hydration product, but only unreacted cement clinker. Fig. 12 presents XRD analysis of initial crack surface hydration products for samples cured 28 d (a) and 210 d (a). The substances of the initial crack surface of 28d mortar were hydration products, calcium hydroxide, and unreacted cement clinker minerals, which provided great potential for crack self-repair. The main purpose is to judge the potential of self-healing ability of substances on the crack surface and analyze various phases on the crack surface

The initial crack surface of the 28 d samples contained calcium hydroxide and unreacted cement clinker minerals, which provided great potential for crack self-repair; whereas the crack surface of the 210-day-old specimens contained generally less calcium hydroxide and unhydrated cement clinker minerals, and the intensity of the diffraction peak of calcium hydroxide gradually decreased with the increase of fly ash admixture and the content decreased.

- Review conclusions after responding to previous comments.

Responses: The conclusion was rewritten.

  1. The increase of fly ash content promoted the improvement of self-repair performance of cracked specimens at 28 d, especially when the fly ash dosage was 40%, the crack opening after 30 days of healing in water was completely closed, the UPV value after recovery was close to 3000 m/s, the self-repair efficiency of maximum amplitude and main frequency amplitude was up to more than 60%, and the recovery rate of compressive strength was increased to know more than 30%. However, the increase of fly ash content was not conducive to the self-repair of cracked samples at 210 d, and with the increase of fly ash content, the crack closure effect weakened, the UPV value after recovery decreased, and the crack repair rate based on ultrasonic transmission decreased to about 20%, and the compressive strength recovery rate increased slightly. The pozzolanic reaction of fly ash improved the self-repair performance of cementitious materials, especially the strength recovery; however, the absence of Ca2+ in fly ash inhibited the precipitation of calcium carbonate, which was not conducive to crack filling.
  2. The initial crack surface of the 28 d samples contained calcium hydroxide and unreacted cement clinker minerals, which provided great potential for crack self-repair; whereas the crack surface of the 210 d samples contained less calcium hydroxide and unhydrated cement clinker minerals, and the intensity of the diffraction peak of calcium hydroxide gradually decreased with the increase of fly ash admixture and the content decreased.
  3. Calcium carbonate precipitation was the main repair product of crack filling and healing, including calcite and spherulite. With the decrease of fly ash admixture, the content of calcium hydroxide increased, which promoted the precipitation of spherulite. With the reaction of fly ash, the precipitation of calcite increased. With the increase of fly ash content, the content of element C in the self-repair products of 28-day-old specimens gradually increased, and the size of calcium carbonate crystals gradually decreased, but the filling was more dense; while the calcium carbonate crystals in the self-repair products of 210d specimens gradually became fine and loose.

Round 2

Reviewer 1 Report

Some answers are superficial. 

Author Response

Review 1

Responses: Thank you very much for your kind letter and the reviewers’ comments. Those comments are very useful for improving our research and revising our manuscript. Based on the suggestions and comments, we have made relevant amendments and we hope the paper is more readable.

2.The author should refine all chapter of the paper.

Responses: Thank you for your suggestions. All chapters of the whole article have been refined and modified under your suggestions, which has increased the integrity, coherence and consistency of the article and improved the readability of the article

  1. Describe the methodology for determining the chemical composition of P.O 42.5 and FA (%).

Responses: According GB/T176-2017, the chemical composition of FA and cement were analyzed by X-ray fluorescence spectrometer (S8 TIGER, Brook). First, a small amount of cement or fly ash powder is pressed into sheets of about 28mm and placed into the instrument. When the chemical elements in the sample are excited by the high-energy radiation emitted by the X-ray tube, they emit characteristic X-rays. Calculate the relationship between mass and the X-ray intensity of an element in the sample through X-ray fluorescence spectrometer, and convert the corresponding element mass percentage, which is listed in Table 1.

  1. How it was determined Fe2O3?

Responses: When the iron element in the sample is excited by the high-energy radiation emitted by the X-ray tube, it emits characteristic X-rays. Calculate the relationship between mass and X-ray intensity of iron in the sample through X-ray fluorescence spectrometer, and convert the corresponding element mass percentage.

  1. How it was determined SO3?

Responses: When the sulfur element in the sample is excited by the high-energy radiation emitted by the X-ray tube, it emits characteristic X-rays. Calculate the relationship between mass and X-ray intensity of sulfur element in the sample through X-ray fluorescence spectrometer, and convert the corresponding element mass percentage.

  1. Describe the methodology for determining the particle size distribution of river sand.

Responses: Put the sieves of different particle sizes, from small to large in order. Put 500g sieve into the largest aperture sieve, put it on the vibrating sieve machine (ZBSX-B2A, Hebwi Shuangxin Co., Ltd.) after vibration, calculate the percentage of sieve on each level of sieve. The results was listed in Table 2.

  1. Justify how the proportions were chosen the “Mix proportion of mortar”- Table 3.

Responses: The water-binder ratio of common concrete is generally about 0.4-0.45, which meets the design standard of C40 (JGJ55—2011). In this paper, a more conventional design standard is chosen to meet most engineering standards.

Table 3

Mix proportion of mortar

No.

Cement /g

Fly ash /g

Water /g

River sand /g

Control

450

0

150

1350

FA-1

405

45

FA-2

360

90

FA-3

315

135

FA-4

270

180

  1. What is the novelty of the paper?

Responses:

Fly ash is a kind of mineral admixture commonly used in concrete. Its ball effect and filling effect play a positive role in improving the working performance and durability of cement-based materials. Moreover, its slow pozzolanic reaction is not only conducive to the development of late strength, but also has a positive effect on the self-healing of cracks. With the dissolution of Ca(OH)2, the pH value in the pore solution increases, resulting in the dissolution of SiO2 and Al2O3 in the fly ash glass phase and the reaction with Ca(OH)2 to produce new hydration products. Although the pozzolanic reaction of fly ash consumes Ca(OH)2 to produce C-S-H gel, fly ash also promotes cement hydration. Both the dissolution of Ca(OH)2 in fractures and the rise of pH value need water environment to achieve. Therefore, this part studies the influence of fly ash content on the self-healing ability of mortar test blocks healed in water, explores the action rule of fly ash content on the self-healing performance of cement-based materials, and provides a theoretical basis for the intrinsic self-healing and durability improvement of cracks of cement-based materials in practical application.

  1. What is the added value of the paper?

Responses:

As a mineral admixture commonly used in concrete, fly ash plays an active role in improving the working performance and durability of cement-based materials. However, the slow reaction of pozzolanic reaction is not only beneficial to the development of late strength, but also has a positive effect on the self-healing of cracks. Therefore, this part discusses the influence of fly ash content on the self-healing ability of mortar test blocks healed in water, studies the action rule of fly ash content on the self-healing performance of cement-based materials, compares the self-healing ability under different ages, and provides a theoretical basis for the intrinsic self-healing and durability improvement of cracks of cement-based materials in practical application.

  1. A comparative analysis between the results obtained and those reported in the specialized literature is necessary. Insert a Table. A scientific discussion is necessary.

Responses:

Table 4 provides a summary and comparison of the cracking and strength recovery achieved through different self-healing mechanisms. Most of the adhesive agents used for self-healing in cementitious materials cannot get hardened in water. By now, almost all types of bacteria explored for self-healing in cementitious materials are the ones that can induce or promote formation of CaCO3. For self-healing based on bacteria the presence of  ions is necessary. For autogenous self-healing, the presence of  ions can enhance the healing efficiency when portlandite in cracks is carbonated. It was reported that by means of autogenous self-healing only the cracks with a width less than 50 μm can be healed. In this study, the use of 40% fly ash in mortar resulted in cross-strength crack healing and strength recovery after 30 days. The fly ash was able to effectively heal cracks up to 270 micrometers with good strength recovery, and was found to be comparable in performance to self-healing concrete incorporating mineral admixtures or bacteria. Overall, the results suggest that the use of fly ash has significant potential for improving the self-healing capabilities of concrete. The self-healing ability of fly ash is not weaker than that of mineral admixtures, bacteria or adhesives.

Table 4

Required conditions for different mechanisms of self-healing.

Mechanisms

Conditions and results

Presence of water in cracks

Presence of  ions in cracks

Upper crack width that can be repaired

Compressive strength

Autogenous

self-healing

In a watery environment

Not necessary. But the presence of  is better.

Less than

50 μm

Compressive strength loss decreases from 27% to 7%[14, 31]

Self-healing based

on mineral

admixtures

In a watery environment

Not necessary.

About 200 μm

Recovering 85%/ (74%) of initial resonant frequency[32]

Self-healing based

on bacteria

In a watery environment

ion is needed

Less than 450 μm

A slight increase in 28 days cured.[33]

Self-healing based on adhesive agents

In most cases cracks should be in a watery environment

Not necessary.

Depending on the amount of agents released

Depending on healing agent [34].

  1. “Figure 16. SEM-EDS analysis of self-healing products at cracks produced in samples cured 28 d.” - In EDS what does "Au" represent.

Responses: "Au" represents the gold. Because the concrete is non-conductive, it is necessary to spray a layer of gold powder on the surface of the sample to increase the electrical conductivity of the sample. In this way, the SEM-EDS pictures of the sample can be better observed. Gold element is added in order to increase the conductivity of the sample when observing SEM, but it is not present in the sample, so it can be seen in SEM-EDS.

Reviewer 3 Report

It is possible to recommend publishing the amended version. All issues raised by the reviewer were correctly addressed by the authors.

Author Response

Thank you for your recognition, we will continue to work hard to write better articles

Reviewer 4 Report

The authors addressed all the comments raised during the reviewing process. Some of the figures have been reconsidered which made the manuscript easier to read and understand.

I, therefore, consider that the manuscript fulfills all requirements to be published.

Author Response

(The authors gave the same response as above.)
